# Prospective One-Year Follow-Up of Sensory Processing in Phelan–McDermid Syndrome

**DOI:** 10.3390/children10061086

**Published:** 2023-06-20

**Authors:** Sergio Serrada-Tejeda, Patricia Sánchez-Herrera-Baeza, Rosa M. Martínez-Piédrola, Nuria Máximo-Bocanegra, Nuria Trugeda-Pedrajo, M.ª Pilar Rodríguez-Pérez, Gemma Fernández-Gómez, Marta Pérez-de-Heredia-Torres

**Affiliations:** Department of Physical Therapy, Occupational Therapy, Rehabilitation and Physical Medicine, Rey Juan Carlos University, Avenida de Atenas s/n, CP 28922 Alcorcón, Spain; sergio.tejeda@urjc.es (S.S.-T.); patricia.sanchezherrera@urjc.es (P.S.-H.-B.); rosa.martinez@urjc.es (R.M.M.-P.); nuria.trugeda@urjc.es (N.T.-P.); gemma.fernandez@urjc.es (G.F.-G.);

**Keywords:** Phelan–McDermid syndrome, *SHANK3*, autism spectrum disorder, sensory reactivity

## Abstract

Background: Phelan–McDermid syndrome (PMS) is caused by the loss (deletion) of a small portion of chromosome 22 in a region designated q13.3 (22q13.3 deletion). PMS is one of the most common genetic forms of autism spectrum disorder (ASD) in which sensory reactivity difficulties have been described on limited occasions. Methods: The objective of this study is to identify whether changes in sensory reactivity skills occur after one year of follow-up in a group of 44 participants diagnosed with PMS. All participants completed the Short Sensory Profile (SSP). Two-factor ANOVA tests were performed with repeated measures for the study of the evolution of the scores. Results: Participants with PMS showed significant changes after one year of follow-up in sensory reactivity skills associated with tactile hyperreactivity (*p* = 0.003). The rest of the study variables did not show significant differences compared to the baseline assessment, showing definite differences associated with patterns of hypo-responsiveness and sensory seeking, low/weak energy, and difficulties in auditory filtering. Conclusions: Understanding the evolution of sensory reactivity skills can facilitate the adjustment to behavioral changes in people with PMS and design-targeted interventions to address sensory reactivity challenges.

## 1. Introduction

A Phelan–McDermid syndrome is a common genetic form caused by haploinsufficiency of the *SHANK3* gene, resulting from a 22q13.33 deletion involving *SHANK3* or a pathogenic variant in *SHANK3* [1,2]. PMS is commonly associated with a diagnosis of autism spectrum disorder (ASD) and is characterized by global developmental delay and severe deficits in language development, in addition to neurological and psychiatric symptomatology [3,4,5].

The variable presentation of the PMS phenotype also includes the presence of signs associated with sensory integrative challenges, including sensory reactivity (i.e., hyper- and hypo-reactivity to sensations, also known as sensory modulation) and sensory seeking. In particular, regulating responses to sensation (sensory reactivity) is important for maintaining optimal states of alertness and focus for attention to tasks. Moreover, behaviors associated with atypical sensory stimuli reflect exaggerated responses and an unusual interest in sensory aspects of the environment, which may limit the performance of activities of daily living as well as social activities and play [6,7,8]. Recently the Autism and Developmental Disabilities Tracking Network (2006–2014), considering the above patterns of sensory reactivity, estimated a prevalence of sensory features at 74% [9]. These sensory traits can manifest varied responses to touch, vision, hearing, taste, smell, and movement. These atypical sensory responses have been collected in the DSM-5 within the Restrictive and Repetitive Behaviors domain for the diagnosis of ASD. These types of behavioral responses have been defined in a new taxonomy proposed by He et al. (2023) [10] under the terms of “Affective responsivity to sensory input” and/or “Behavioural responsivity to sensory input”, with the aim of reflecting the affective or behavioral sensory responses.

The current literature has mainly described these sensory reactivity difficulties in idiopathic autism [7,11,12,13,14]. Different studies have identified that sensory processing difficulties can have a negative impact on daily life [6], and although a clear relationship between the two has not yet been established, different cross-sectional studies have identified that sensory profiles associated with increased sensory seeking are associated with poorer communication skills and participation in daily activities [15]. Similarly, longitudinal studies have identified that during infant development, the presence of sensory hyporreactivity patterns appeared to negatively impact long-term social and communication skills [16], while sensory hyperreactivity profiles have been associated with poorer abilities to participate in activities of daily living [17].

However, these sensory profiles have also been described in individuals with a genetic etiology underlying an initial diagnosis of ASD or intellectual disability (ID) [17,18,19,20,21]. In the case of PMS, atypical sensory responses are characterized by a higher identification of signs of sensory hyporreactivity, especially a reduced pain response and differences in auditory information processing. In addition, signs of tactile hyperreactivity appear to be common in all studies conducted, as well as fewer signs of auditory hyperreactivity and sensory seeking [1,3,20,21,22,23].

With advances in research, sensory processing difficulties have become increasingly evident and have acquired greater relevance in the scientific arena. However, detailed information on their evolution in individuals with autistic symptomatology is limited. Available studies suggest that sensory processing traits remain stable during childhood in individuals with ASD [24,25], and a reduction in severity appears to be observed over the years [26]. Similar results have been reported in a recent study [27], in which stability was observed in scores on the SSP reporting little change during early childhood. Although the contributions of previous research report on the stability of sensory features in persons with ASD, the results are inconclusive due to different aspects, such as sample sizes or the lack of differentiation between specific types of sensory response patterns.

Similarly, little scientific literature available reporting on the evolution of sensory traits in PMS is available. Currently, the only four longitudinal studies available on the PMS population [28,29,30] report that, although the PMS population acquires some basic skills, improvement or enhancement of these skills is very limited, probably due to the process of regression and stagnation, that occurs throughout the evolution of the disease. In addition, progressive regression is observed in developmental skills. In this sense, language development skills, especially expressive language, was the area that showed more severe impairment compared to receptive language skills or gross motor function, which seem to show better development. These findings have also been identified in other observational studies that have characterized the regression process in SMP [31,32,33], but the developmental reported regression has not been fully characterized and defined. However, limited information on the evolution of sensory processing skills was reported. In this respect, Philippe et al. [34] conducted a four-year follow-up in eight children with single 22q13.3 deletion and, similar to the current studies [20,22], all children had sensory processing abnormalities associated with atypical sensory seeking responses, hypo-reactivity, as well as signs of tactile hyper-reactivity. However, although they reported that sensory features appear to reduce with developmental age, the data should be interpreted carefully, as they did not use a specific sensory assessment for sensory reactivity.

Considering that the symptomatology associated with sensory processing may be an aspect that precedes the appearance and identification of the central features of the ASD diagnosis, such as difficulties in social communication or the presence of repetitive and restricted behaviors, it is necessary to understand the evolution of sensory features [35]. Therefore, and given the scarce availability of studies, the aim of the present research is to conduct a longitudinal study in the PMS population to identify whether sensory processing skills related to sensory reactivity respects vary over a one-year period.

## 2. Materials and Methods

### 2.1. Study Design

A one-year longitudinal cross-sectional study was designed. The study was approved by the ethics committee of the Universidad Rey Juan Carlos. The participants’ families completed the informed consent document and provided the genetic reports necessary for diagnostic confirmation. This project was initially conducted in Spain (July 2020). Data management was performed in accordance with the criteria established in the Declaration of Helsinki [36], the Data Protection Regulation (EU) 2016/679 [37], and the Spanish legal framework in force regarding personal data protection [38].

### 2.2. Participants

A group of 44 patients diagnosed with PMS was recruited. Participants were initially recruited in January 2021 and were followed up for one year after the first evaluation (Figure 1). All participants who were recruited for the study belong to the Phelan–McDermid Syndrome Association of Spain. To participate in the study, participants provided the necessary documentation for diagnostic confirmation. In the case of a diagnosis associated with a *SHANK3* deletion, a comparative genomic hybridization (CGH) test was required. In the case of *SHANK3* gene mutation or sequence variants, whole exome sequencing (WES) analysis was required.

Given the presence of comorbidity of ASD diagnosis in the PMS population, it was considered appropriate to include the participants with a confirmed diagnosis and/or features of ASD as described in the DSM-5 criteria.

### 2.3. Procedure

For the follow-up of the participants in the first phase of the study [23], the Spanish Short Sensory Profile (SSP-S) [39] was administered after the first assessment was completed. This questionnaire was completed by the primary caregiver. When necessary, if the participants identified difficulties in understanding the questionnaire items, the research team provided telephone assistance to those who requested help. Specifically, four primary caregivers were contacted to resolve doubts related to understanding the instructions. In addition to the questionnaire, a document prepared by the research team was sent to collect information regarding the participant, sociodemographic data, type of genetic disorder, and attendance to rehabilitation treatments.

### 2.4. Variables and Data Measurements

Sociodemographic and genetic data were recorded for each participant, including age, gender, type of genetic alteration, primary caregiver, and use of rehabilitation resources. The SSP-S scores of each participant were collected at one year of follow-up. As in the first phase of the study, the culturally adapted version of the Short Sensory Profile Spanish (SSP-S) was used [39,40].

The SSP is a common sensory feature screening instrument based on the Sensory Profile, a questionnaire designed by Dunn et al. [41]. It consists of 38 items organized into seven subcategories that inform about the aspects related to tactile sensitivity, taste/smell sensitivity, movement sensitivity, underresponsive/seeks sensation, auditory filtering, low/weak energy, and visual/auditory sensitivity. For the score, a Likert scale from 1 to 5 reflects the frequency of behaviors identified by caregivers (1, always; 2, often; 3, sometimes; 4, almost never; 5, never). The raw score for each subcategory is associated with a threshold that allows different categories to be established: typical performance, probable difference (1 SD below the mean), and definite difference (2 SD below the mean). The greater the presence of behaviors related to sensory processing difficulties, the lower the scores. The SSP total score and subcategories scores are based on the percentiles of a typically developed sample of children (Table 1).

### 2.5. Data Analysis

For the descriptive statistical analysis of the sample, the basic descriptive methods were used. For qualitative variables, the number of cases present in each category and the corresponding percentage were calculated, and for quantitative variables, the mean and standard deviation were calculated.

For the study of the evolution of the scores on the SSP-S scale, two-factor ANOVA tests were performed with repeated measures using the General Linear Model procedure. Two-to-two comparisons were performed using the Bonferroni adjustment. The analysis of the variables was performed with the statistical program IBM SPSS Statistics for Windows, V.27.0 (Copyright 2013 IBM SPSS Corp., Chicago, IL, USA). The differences considered statistically significant are those whose *p* < 0.05.

## 3. Results

### 3.1. Socio-Demographic and Genetic Description of the Sample

Fifty-one families from the Phelan–McDermid Syndrome Association of Spain were contacted (Table 1). However, seven participants dropped out of the study because they did not have time to complete the evaluations or did not want to continue the study. A total of 44 participants with PMS (86.2% response rate) were finally recruited, consisting of 22 males and 22 females, with a mean age of 12.6 years (SD 7.80). In 38 cases (86.4%), a *SHANK3* deletion was identified. The deletion size presented a high variability, ranging from 52 kb to 8.53 Mb. *SHANK3* mutation was confirmed in the remaining six participants (13.6%). Table 1 shows the sociodemographic and genetic information. In addition, to understand the results of the sensory reactivity evaluation, the score intervals of each sensory quadrant were included, as well as the total SSP-S score.

### 3.2. Sensory Reactivity Scores according to the PMS Genetic Alteration

Table 2 shows the distribution of scores obtained in the SSP-S according to the type of genetic alteration identified in the sample of participants. In the case of the participants with *SHANK3* deletion, an atypical sensory reactivity profile associated with a definite difference (62.2%) is mostly observed, in contrast to the sensory profile of the participants with *SHANK3* mutation, who show a more variable sensory profile.

In relation to the sensory quadrants, in the case of *SHANK3* deletion participants, a higher frequency of responses associated with definite differences is observed in the Tactile Sensitivity (40.5%), Underresponsive/Seeks Sensation (67.6%), Sensory reactivity difficulties associated with Auditory Filtering (54.1%), and Low/weak energy (73%) sections. In contrast, the sample of *SHANK3* mutation participants shows sensory reactivity profiles in which responses associated with a lower presence of atypical sensory reactivity responses are observed. In both samples of participants, it is observed that the sensory reactivity skills of the taste/smell sensitivity category, as well as the visual/auditory sensitivity section show a typical performance in a very high percentage of the participants.

### 3.3. Comparative One-Year Follow-Up Analysis

The descriptive and comparative analysis of the variable scores before and after treatment is shown in Table 3. The results showed that tactile sensitivity increased significantly at the end of treatment with respect to baseline (*p* = 0.033) (Figure 1). The significant change in tactile sensory reactivity skills implies a change in the category of sensory processing difficulties. In this case, the baseline score corresponds to a score within the interval associated with a definite difference (Tactile Sensitivity score: 7–26), versus the score at one year of follow-up, with scores within the interval indicating a probable difference (Tactile Sensitivity score: 27–29). No statistically significant changes were observed in the rest of the variables (Figure 2).

## 4. Discussion

The identification of changes in the sensory profile in a sample of people diagnosed with PMS in the Spanish population provides information on sensory reactivity skills, currently referred to as affective and/or behavioral responses to sensory input.

As in previous studies, difficulties associated with sensory reactivity have been identified when assessing sensory processing skills in the PMS population [3,21,22,23,42]. In this case, our results report that approximately 76% of participants with PMS show sensory reactivity difficulties, which are associated with a definite or probable sensory difference. In this case, the atypical sensory profile is associated with behavioral responses that reflect a pattern of underresponsiveness/sensory seeking, low/weak energy, auditory filtering, and tactile sensitivity sensory challenges.

In this sense, our results are consistent with those obtained in previous studies that have employed the SSP in the PMS population [22,43,44], reflecting a high frequency of these types of behavioral responses. With regard to the pattern of sensory responses of hyporreactivity and seeking sensation, this has been characterized by a high sensory threshold associated with both passive responses to sensory stimuli (hypo-reactivity) and active sensory-seeking patterns. In line with our results, the studies of Mieses et al. [22] and Droogmans et al. [42] identified an atypical sensory profile associated with a definite difference in the low/weak energy quadrant. These manifestations have been linked to hyporreactivity of the vestibular and proprioceptive systems, which may be related to the presence of signs such as hypotonia or motor difficulties observed in PMS.

On the other hand, and in contrast with the studies that have employed the SSP, Tavassoli et al. [21], using The Sensory Assessment for Neurodevelopmental Disorders (SAND), identified a specific sensory phenotype of PMS characterized by a higher presence of signs of sensory hyporreactivity and a lower presence of signs of auditory sensory search. This phenotype is clearly different from that identified in idiopathic ASD. In comparison with the SSP, the SAND clearly differentiates the type of behavioral response associated with a hypo-reactive or sensory-seeking sensory profile, which facilitates the interpretation of the results. In this case, Tavassoli et al. [21] identified high percentages of hyporreactivity and sensory seeking (92% and 65%, respectively).

On the other hand, the results of our study have identified that the tactile sensory reactivity skills of the sample of participants with SSP are the only ones that showed significant changes after one year of follow-up. Compared to our previous results [23], a significant qualitative change has been identified in the signs associated with tactile sensitivity, mostly signs of hyperreactive behavioral responses. In this case, the changes indicate a probable, rather than definite, difference in participants with PMS. In this sense, the change in score is associated with a reduction in the level of alertness and the appearance of behavioral responses.

Although a change in these sensory reactivity skills has been observed, previous studies have identified the existence of a profile of tactile sensory hyperreactivity in all ages, including children [34,45]. However, although only significant in one sensory modality, stagnation or slight changes in scores are observed in the rest of the sensory sections assessed, which, although not significant, suggest that sensory reactivity skills appear to remain stable over a short period of time.

In the PMS population, only one longitudinal study has been performed [34], in which skills related to sensory processing seem to be reduced. However, the test used for the assessment is not sensory-reactivity-specific, which may limit the interpretability of the results, given the importance of selecting appropriate tests, such as the SSP or the SAND, for the measurement of those sensory difficulties. In contrast, studies in children with ASD [24,25,26,27] have identified that there are no substantial changes in SSP scores, and although they are often associated with atypical sensory processing, these scores remain stable over time. This stability in the scores may be associated with the evolution of the disease. In this sense, different stages or phases of regression in their developmental abilities have been identified in people with PMS, after which periods of rest follow. In this case, the study by Dille et al. [33] proposes a conceptual model that identifies four differentiated neurodevelopmental regression stages to facilitate the monitoring of the population with PMS. In our case, the average age of the participants was 12 years, which, according to the previous classification, would place the participants in a stagnation phase that would begin approximately at 8 (after the first regression) and that could last up to 10 years.

Considering the biological mechanisms underlying sensory processing, *SHANK3* is known to affect glutamatergic processing and encodes for a protein that connects membrane receptors to actin in the cytoskeleton, thereby providing stability and integrity to the synaptic structure [46]. However, although the literature has identified that deficiencies in the *SHANK3* gene can affect glutamatergic processing, its relationship to sensory processing, especially hyporreactivity, is not entirely clear [7,47,48]. On the other hand, the signs of tactile sensory reactivity are a common feature in PMS, and the biological basis for these tactile sensory reactivity difficulties has been identified in previous studies that have established a relationship between GABAergic activity and sensory reactivity. For example, Sapey-Triomphe et al. [49] identified a positive relationship between signs of tactile sensory hyperreactivity in adults with ASD and further suggested that the alterations in GABA modulation observed in ASD are related to increased difficulties in predicting tactile input. Similarly, the study by Tavasolli et al. [50] identified that in typically developing children, variants in the *GABRB3* gene were significantly associated with tactile sensory reactivity.

Given the limited sample size of the participants, the results of our study should be interpreted with caution. Larger follow-up studies would be advisable to identify the evolution of sensory processing skills more comprehensively. Although our results provide objective data on sensory reactivity difficulties, this study has essentially relied on the use of parent reports to describe how children with ASD react to different stimuli presented in various contexts, so the use of observation-based tests, such as the SAND, would identify signs of sensory difficulties in the child’s sensory processing skills based on the DSM-5 criteria for ASD. In addition, it would be advisable to carry out longitudinal studies that would identify the progression of reactivity and sensory processing skills in people with PMS who are in a stage of neuropsychiatric decompensation to facilitate the understanding of the appearance of behaviors that may be linked to sensory factors.

## 5. Conclusions

For the first time, a longitudinal follow-up of sensory reactivity abilities in the PMS population has been performed using a specific sensory assessment test. Overall, our results suggest that tactile sensory reactivity abilities seem to suffer a slight improvement in comparison to the rest of the variables, which remain stable. Our results support the importance of designing targeted interventions to address sensory integrative challenges, which is consistent with recommendations established by an international consortium of experts and patient representatives [51] that suggest considering sensory reactivity difficulties in the presence of behavioral changes in the person with PMS. Although the SSP allows the identification of behavioral responses associated with sensory reactivity difficulties, considering that the autistic population presents not only sensory reactivity or sensory seeking difficulties, the consideration and evaluation of other sensory perceptual aspects and sensorimotor skills, such as sensory perception or praxis, should be considered.

## Figures and Tables

**Figure 1 children-10-01086-f001:**
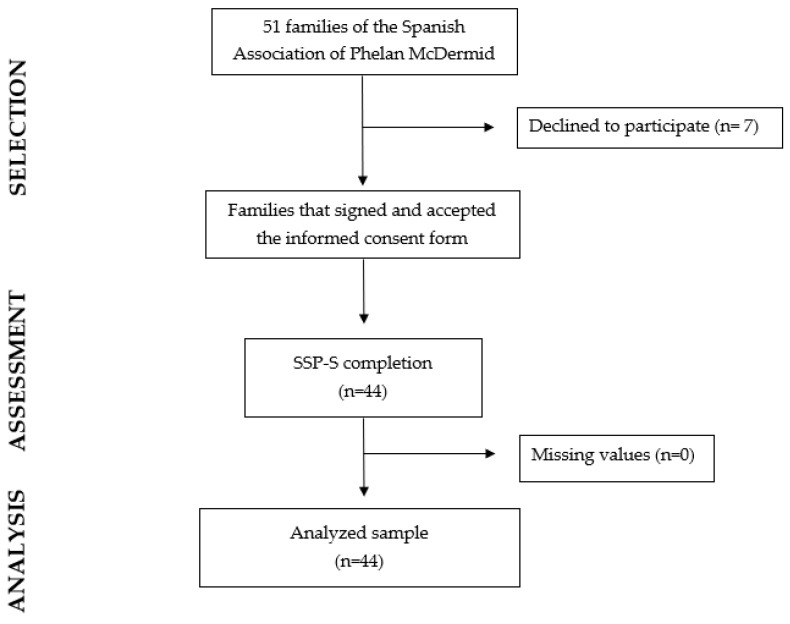
Flow diagram.

**Figure 2 children-10-01086-f002:**
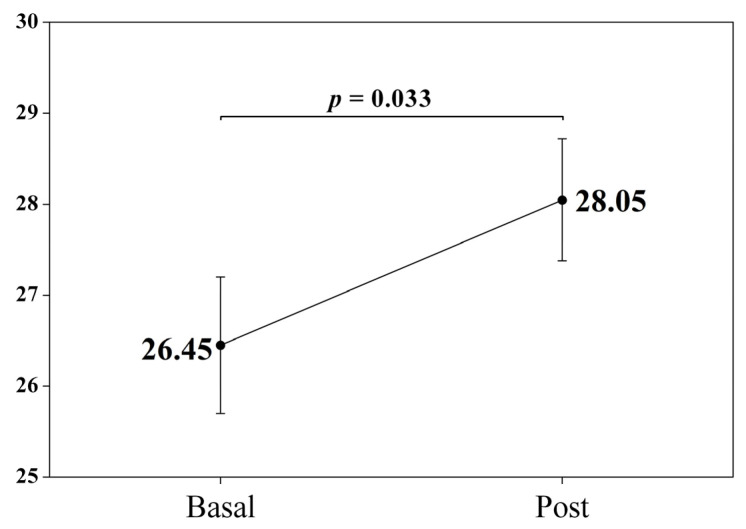
Evolution of tactile sensitivity scores in PMS after one year of follow-up.

**Table 1 children-10-01086-t001:** Sociodemographic data and sensory profile score classification (SSP-S) (adapted from Dunn).

**Age (years), mean (SD)**	12.6 (7.80)		
**Age range (years), *n* (%)**			
From 3 to 5	3 (7%)		
From 6 to 11	25 (57%)		
From 12 to 17	8 (18%)		
From 18 to 24	4 (9%)		
from25 to 40	4 (9%)		
**Gender. *n* (%)**			
Male	22 (50%)		
Female	22 (50%)		
**Genetic alteration, *n* (%)**			
Deletion	38 (86%)		
Deletion size interval	52 kb–8.53 Mb		
Point mutation	6 (14%)		
**Caregiver, *n* (%)**			
Mother	22 (50%)		
Father	1 (2%)		
Both parents	20 (46%)		
Another person	1 (2%)		
**Rehabilitation services, *n* (%)**			
Physical therapy	23 (52.3%)		
Speech and language therapy	34 (77.3%)		
Psychology	19 (43.2%)		
Occupational therapy	10 (22.7%)		
**SSP Section/Subscale (items)**	**Classification**		
	Definite Difference	Probable Difference	Typical Performance
Tactile Sensitivity (1–7)	7–26	27–29	30–35
Taste/Smell Sensitivity (8–11)	4–11	12–14	15–20
Movement Sensitivity (12–14)	3–10	11–12	13–15
Underresponsive/Seeks Sensation (15–21)	7–23	24–26	27–35
Auditory Filtering (22–27)	6–19	20–22	23–30
Low/Weak Energy (28–33)	6–23	24–25	26–30
Visual/Auditory Sensitivity (34–38)	5–15	16–18	19–25
Total score (1–38)	38–141	142–154	155–190

Note: SD: standard deviation; SSP-S: Short Sensory Profile—Spanish; SSP-S’ classification is based on the developed children (*n* = 1037). No missing data were reported.

**Table 2 children-10-01086-t002:** Descriptive statistics of sensory reactivity in PMS population.

	*SHANK3* Deletion (*n* = 38)				*SHANK3* Mutation (*n* = 6)			
	Score Median (IQR)	DD N (%)	PD N (%)	TP N (%)	Score Median (IQR)	DD N (%)	PD N (%)	TP N (%)
Total score	136 (77–173)	24 (62.2%)	5 (13.5%)	9 (24.3%)	147.50 (116–169)	2 (33%)	2 (33%)	2 (33%)
Tactile Sensitivity	28 (18–35)	16 (40.5%)	5 (13.5%)	17 (45.9%)	30 (21–32)	2 (33%)	---	4 (66%)
Taste/Smell Sensitivity	20 (4–21)	1 (2.7%)	3 (5.4%)	34 (91.9%)	20 (16–20)	---	---	6 (100%)
Movement Sensitivity	13 (5–15)	10 (24.3%)	5 (13.5%)	23 (62.5%)	13 (8–15)	2 (33%)	---	4 (66%)
Underresponsive/Seeks Sensation	21 (8–32)	26 (67.6%)	5 (13.5%)	7 (18.9%)	25 (17–32)	2 (33%)	2 (33%)	2 (33%)
Auditory Filtering	19 87–27)	21 (54.1%)	11 (29.7%)	6 (16.2%)	21.50 (17–27)	3 (50%)	---	3 (50%)
Low/ Weak Energy	17 (6–30)	28 (73%)	1 (2.7%)	9 (24.3%)	21 (8–33)	3 (50%)	1 (16%)	2 (33%)
Visual/Auditory Sensitivity	20 (8–25)	6 (16.2%)	5 (13.5%)	27 (70.3%)	20 (16–23)	---	2 (33%)	4 (66%)

Note: DD, definite difference; PD, probable difference; TP, typical performance; IQR, interquartile range; SD, standard deviation; SSP-S, Short Sensory Profile—Spanish.

**Table 3 children-10-01086-t003:** Comparative analysis of measurements at baseline and at one-year follow-up in PMS population.

	Measure, Mean (SD)		Mean. Dif.	Student-*t* Test		*d*
	Pre	Post		*t* (43)	*p*-Value	
**Total score** (1–38)	133.86 (19.65)	137.84 (20.96)	−3.98	−1.76	0.086	−0.27
Tactile Sensitivity (1–7)	26.45 (4.96)	28.05 (4.43)	−1.59	−2.20	**0.033 ***	−0.33
Taste/Smell Sensitivity (8–11)	18.20 (2.95)	18.48 (3.00)	−0.27	−0.88	0.382	−0.13
Movement Sensitivity (12–14)	12.39 (3.01)	12.55 (2.61)	−0.16	−0.44	0.66	−0.07
Underresponsive/Seeks Sensation (15–21)	20.93 (6.63)	21.16 (6.02)	−0.23	−0.32	0.749	−0.05
Auditory Filtering (22–27)	18.64 (4.90)	19.20 (4.61)	−0.57	−0.93	0.359	−0.14
Low/Weak Energy (28–33)	18.41 (6.86)	18.98 (7.12)	−0.57	−0.91	0.369	−0.15
Visual/Auditory Sensitivity (34–38)	19.43 (3.49)	19.43 (3.78)	0.00	0.00	1	0

Note: *d*: Cohen’s d (effect size). * Significant at *p* < 0.05.

## Data Availability

The data are available on request from the corresponding author. The data are not publicly available due to privacy aspects.

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
