# Peer review of "Prospective One-Year Follow-Up of Sensory Processing in Phelan–McDermid Syndrome"

_children, 2023, doi:10.3390/children10061086_

Round 1

Reviewer 1 Report

This study contributes new information on sensory processing issues in Phelan-McDermid syndrome using a caregiver questionnaire. I do not think the statistical analysis showing a change in tactile sensitivity is very convincing. However, the data collected on these patients' specific sensory issues after 1 year will be a useful contribution to the literature since it helps confirm a sensory phenotype of Phelan-McDermid syndrome and improves understanding of this rare genetic disorder.

Comments:

1) Was the statistical analysis corrected for multiple comparisons using Bonferroni or other correction? If not, please redo it with the appropriate correction and explicitly state it in the methods, or comment on why it does not apply.

2) After multiple comparison correction I suspect the change in tactile sensory reactivity will not be significant. Is there some reason the authors expect a change in this specific area in response to treatment (or progression over time)? If so, more explanation about that would be helpful. Otherwise, it may be better to just present that (as the authors already discuss) sensory issues were stable overall in the patient population for a year.

Language is mostly appropriate but may need some minor fixes.

Please consider italicizing SHANK3 (the gene name) following HGNC guidelines

introduction: "global developmental delay" is a more standard term than "general developmental delay"

introduction: "However, limited on the evolution of sensory processing skills were reported". Do you mean to say 'limited information'?

Author Response

Thank you very much for your response. Research on the PMS population has increased in recent years due to the interest of the scientific community in understanding the functional difficulties of this population. Although most studies focus on genetic analysis and the molecular basis of the deletion or mutation, more and more information is needed with more objective clinical evaluation tests.

We welcome your comments and believe that the analysis procedure is appropriately tailored. We personally believe that possible doubts about the results may be due to the fact that in the initial version of the manuscript, we omitted the specific section on the type of analysis performed. We hope that after the changes made, everything will be clearer.

Was the statistical analysis corrected for multiple comparisons using Bonferroni or other correction? If not, please redo it with the appropriate correction and explicitly state it in the methods, or comment on why it does not apply.

Response: Thank you very much for your comment. It was our mistake in the drafting of the manuscript, since in the initial version we did not include the statistical analysis section and therefore we did not indicate the analysis procedure performed.

However, we have contacted the statistical analysis team again to confirm the procedure performed and the results reported in the first version are correct, since the Bonferroni adjustment was performed in the same analysis.

Lines 128-136: For the descriptive statistical analysis of the sample, the basic descriptive methods were used: for qualitative variables, the number of cases present in each category and the corresponding percentage were calculated, and for quantitative variables, the mean and standard deviation were calculated.

For the study of the evolution of the scores on the IRI and JSE scales, two-factor ANOVA tests were performed with repeated measures in one of them using the General Linear Model procedure. Two-to-two comparisons were performed using the Bonferroni adjustment. Statistical analysis was performed with SPSS 25.0 for Windows. The differences considered statistically significant are those whose p < 0.05.

After multiple comparison correction I suspect the change in tactile sensory reactivity will not be significant. Is there some reason the authors expect a change in this specific area in response to treatment (or progression over time)? If so, more explanation about that would be helpful. Otherwise, it may be better to just present that (as the authors already discuss) sensory issues were stable overall in the patient population for a year.

Response: Thank you very much for your comment. As in the previous manuscript, we believe that the doubts about the interpretation are due to the omission of the corresponding statistical analysis section, so that by not having indicated the type of procedure, the results could appear doubtful.

However, the information available after the review facilitates the interpretation and understanding of the results, which, as indicated in the analysis performed, already included the Bonferroni correction in their procedure.

Lines 128-136: For the descriptive statistical analysis of the sample, the basic descriptive methods were used: for qualitative variables, the number of cases present in each category and the corresponding percentage were calculated, and for quantitative variables, the mean and standard deviation were calculated.

For the study of the evolution of the scores on the IRI and JSE scales, two-factor ANOVA tests were performed with repeated measures in one of them using the General Linear Model procedure. Two-to-two comparisons were performed using the Bonferroni adjustment. Statistical analysis was performed with SPSS 25.0 for Windows. The differences considered statistically significant are those whose p < 0.05.

Please consider italicizing SHANK3 (the gene name) following HGNC guidelines

Response: Thank you very much for your comment. We fully agree with your appreciation. It was a formatting error during the writing process. We have made the changes you requested.

Lines 24; 25; 85; 86; 145; 146; 174: SHANK3

Introduction: "global developmental delay" is a more standard term than "general developmental delay"

Response:  Thank you very much for your comment. We have made the change as your suggestion is well suited to the terminology currently in use.

Lines 26:  […] global developmental delay […]

Introduction: "However, limited on the evolution of sensory processing skills were reported". Do you mean to say 'limited information'?

Response: Thank you very much for your comment. We have made the modification since it was a mistake in the wording. 

Lines 63-64: However, limited information on the evolution of sensory processing skills were reported

Reviewer 2 Report

Study Design

This study was conducted in Spain, ........''authors should state the period/Month & year of the study.

In my opinion given the sparsity of data in this area of research, the present study has added to thefund of knowledge. Authors have also carefully acknowledged the limitation of their study and sounded cautionary in the interpretation of their result in view of the limitation.

Author Response

Thank you very much for your comment. We are aware of the limitations of the study. However, given the scarce information, and the importance that sensory reactivity difficulties have in the diagnosis and understanding of autistic symptomatology, we consider the findings to be useful.

This study was conducted in Spain:

''authors should state the period/Month & year of the study.

Response: Thank you very much for your comment. We will proceed to make the adjustments in the indicated section.

Line 75: This project was initially conducted in Spain (July/2020)